# Exploring the Missing Link between Life Cycle Assessment and Circularity Assessment in the Built Environment

Wai Chung Lam [1,2,*] , Steven Claes [1,2] and Michiel Ritzen [1,2]

1    VITO, Unit Smart Energy and Built Environment, Boeretang 200, 2400 Mol, Belgium
2    EnergyVille, Thor Park 8310, 3600 Genk, Belgium
*    Correspondence: waichung.lam@vito.be

**Abstract:** The built environment—with its significant environmental impact and long lifespan—is a key sector in which to implement circular economy principles. So far, however, a coherent framework with circularity indicators has not yet been established. While Life Cycle Assessment (LCA) is commonly practiced to quantify the total environmental impact, it is designed and intended to assess linear life cycles, making it unsuitable for circularity assessment. Thus this paper explores an objective method to link LCA to a semi-quantified circularity indicator. Four variants of external wall designs of two case studies are analyzed. The explored possibilities of linking LCA results or mass input flows to the circularity indicator show differences in outcome. Furthermore, the analysis shows that decision-making can be affected by choice of assessing only a circularity indicator, an environmental impact indicator, or mass input flows, as opposed to a scenario in which a linked approach of these parameters is applied.

**Keywords:** Life Cycle Assessment; circularity assessment; linking indicators

## 1. Introduction

### 1.1. The European Policy Context

Buildings are responsible for about 50% of resource extraction and consumption, over 30% of the European Union's yearly total waste, 40% of the European Union's energy consumption, and 36% of the energy-related greenhouse gas emissions [1]. To address these issues, the European Commission (EC) has published the revised Construction Products Regulation (CPR) and Sustainable Products Initiative (SPI).

The goal of the revision of the CPR is to create a harmonized framework to assess and communicate the environmental and climate performance of construction products. New product requirements are included to make construction products more durable, repairable, recyclable, and easier to remanufacture [1]. However, when referring to the proposal for the revised CPR [2], there is no mention of a clear method to characterize inherent product requirements related to a circular economy. With the SPI, the Ecodesign Directive will be revised, and additional legislative measures will be proposed [3]. The proposed revision of the Ecodesign Directive [4] mentions that the EC should continue efforts to develop and improve science-based assessment tools. Furthermore, it states that the EC should build further on existing tools, like the Product Environmental Footprint (PEF) method, and use dedicated studies when needed to further reinforce circularity aspects in the assessment of products and in the preparation of ecodesign requirements.

To enable the uptake and measure of the actual effects of circular principles, a clear, quantifiable method should be applied. A scientific framework to measure the level of circularity of different building solutions (i.e., in this paper: elements, components, materials, etc.—all levels smaller than building level) could lead to a broad understanding of circular building practices and—as a result—to a more resilient built environment [5]. So far, however, no single widely-accepted definition of circularity principles has been

established [6–8]. Hence, a harmonized, coherent set of quantifiable indicators—suitable to measure circularity aspects of building solutions—is still to be fully developed.

### 1.2. The State of the Art of Circularity Indicators

Several frameworks regarding circularity principles exist. For instance, Saidani et al. [9] classified 55 circularity indicators based on criteria such as the implementation level (micro, meso or macro), the type of loops (maintain, remanufacture/reuse or recycle), and possible purposes (informative, action-oriented, communicative or educational). The majority of the 55 identified circularity indicators are non-sector-specific. Only the Building Circularity Index (BCI) [10] was included as an indicator tailored to the construction sector.

Concerning construction sector-specific indicators, Cambier et al. [8] identified 38 design support tools for circular building relevant to the Belgian construction sector—tools ranging from circular design strategy frameworks aiming to offer guidance without quantifying the circularity (e.g., the design for change design guidelines of the Flemish Public Waste Agency (OVAM) [11] or the design quality guide by Vrije Universiteit Brussel (VUB) [12]), to circularity scoring tools aiming to objectify the circular performance of buildings or building solutions via a scoring or assessment system on one single or multiple circularity indicators (such as the BCI [10], Reuse Potential Tool (RPT) [13], Transformation Capacity Tool (TCT) [14], Circulytics [15], or the Circular Transition Indicators (CTI) [16]). An international standard also exists, i.e., the ISO 20887:2020 [17], which provides an overview of design principles for disassembly and adaptability, and includes how to measure the performance of thirteen principles. Moreover, the EU has established the Level(s) framework, consisting of six macro-objectives, of which the second macro-objective is called "Resource efficient and circular material life cycles". Two of the four indicators under that second macro-objective relate to assessing the material and waste flows, and the other two to how design can facilitate future adaptation and deconstruction [18].

The examples of circularity frameworks and tools mentioned above are not coordinated. Moreover, the multitude of definitions and indicators regarding circularity in the built environment does not contribute to a coherent, systematic approach, leading to divergent interpretations and results [6]. Thus, the need exists for a coordinated set of objectively quantifiable circularity indicators in order to move towards a sustainable, circular built environment.

### 1.3. The State of the Art of Quantifying the Environmental Impact of Circular Buildings

When it comes to quantifying the total environmental impact of buildings and buildings solutions, Life Cycle Assessment (LCA) is common practice [19–22]. However, the current European standards on LCA of construction products and buildings, i.e., EN 15804+A2:2019 [23] and EN 15978:2011 [24], respectively, are intended to assess linear life cycles and not multiple or circular life cycles. The standard EN 15978, for instance, does not cover impacts related to the potential benefits of Design for Disassembly (DfD) [25]. In light of this shortcoming, we propose a method to model the flows of building parts initiated by the disassembly of a building. Circular LCA and MFA approaches are currently not included in LCA tools, as implementing such an approach would require adapting the LCA scope, the calculation method, and the datasets used by the construction industry [26].One of the possibilities to overcome this linearity issue would be to apply a different allocation approach, such as the Circular Footprint Formula of the PEF method by the EC [27], the linearly degressive approach [20] or the Circular Building Life Cycle Assessment [28]. Another possibility would be to incorporate time-based case-by-case specific refurbishment and/or replacement scenarios in a comparative LCA between the circular solution(s) and the representative business-as-usual reference scenario [25,28]. However, results can vary largely from one assumed user scenario to the other [28].

### 1.4. The Missing Link between LCA and Circularity Assessments

Current quantitative circularity assessments are often based on material flow analysis (MFA), expressing results in masses of materials or (mass) percentages of the total amount of

assessed materials, water and energy flows (e.g., Circulytics [15], the CTI [16], the CE-LCA [29] and the method developed by González et al. [30]). Qualitative and semi-quantitative assessment frameworks involve subjective valuations or simplified rating schemes of a certain selection of circularity indicators (e.g., RPT [13] and TCT [14]). The assessment of the potential contribution to environmental impact categories as in LCA forms no part of the circularity assessment frameworks mentioned above. Adoption of LCA can contribute to the comprehensiveness and transparency of circular solutions in the construction industry, but there is no recognized circular economy assessment framework for the construction industry [31]. In the study by Rajagopalan et al., a circular building case study was assessed with a multi-criteria decision analysis by combining a quantitative LCA-based method with a set of qualitative circularity criteria. A drawback of this multi-criteria decision analysis, however, is the requirement of a weighting method to arrive at a singular choice at the end of the analysis [28]. The weighting was not included in their study, as they deemed it to be subjective and dependent on the preference of the decision-maker.

In order to reduce subjectivity in circularity assessments and assumptions related to case-specific scenarios that can influence the results of comparative LCA of circular building solutions, this study aims to explore the possibility of linking LCA to a circularity indicator. The aim is to investigate the effects of a simple objective method to assess the potential environmental impact and the circularity of building solutions expressed in one single score. It is an exploration of integrating LCA and a circularity assessment (as visualized in Figure 1) instead of performing them in parallel like done in current practice. This study, therefore, attempts to bridge the gap between quantitative LCA outcomes and a dimensionless simplified circularity rating. The need to clarify the link between LCA and circularity indicators is also underlined by the set-up of a joint research group between the Society for Environmental Toxicology And Chemistry (SETAC) and the American Center for Life Cycle Assessment (ACLA) [32].

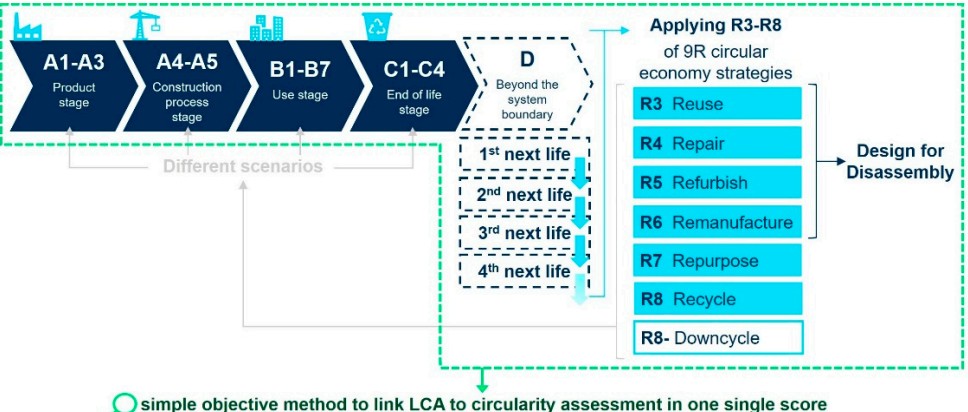

**Figure 1.** The aim of this paper: linking LCA (left part of the figure based on EN 15804 [23]) to circularity assessment (assessing the application of circular economy strategies as visualized in the right part of the figure) in one single score without the need to define case-specific scenarios.

This study takes detachability based on the type of connection as the integrated circularity indicator within the linked method (which will be further explained in Section 2.2.2), as it is the least dependent on the interpretation of the assessor, and it is the first step in the hierarchy of assessing the reversibility of a circular solution (as shown in Figure 2). If a solution cannot be detached without demolishing it—e.g., in the case of chemically bonded connections—there is no ground to assess other circularity aspects, such as accessibility and ease of disassembly. In addition, it is an essential property for determining the environmental impact of possible reuse and thus influences the definition of end-of-life (EOL) scenarios in LCA. Furthermore, LCA is applied next to MFA to assess whether mass input flows based on MFA and LCA, resulting in potential environmental impact, would lead to a different choice of building solutions. The

linked methodology is applied to two case studies, each consisting of four variants of external wall compositions, to test what the effect of linking LCA or MFA to detachability is and if the way in which they are linked is important.

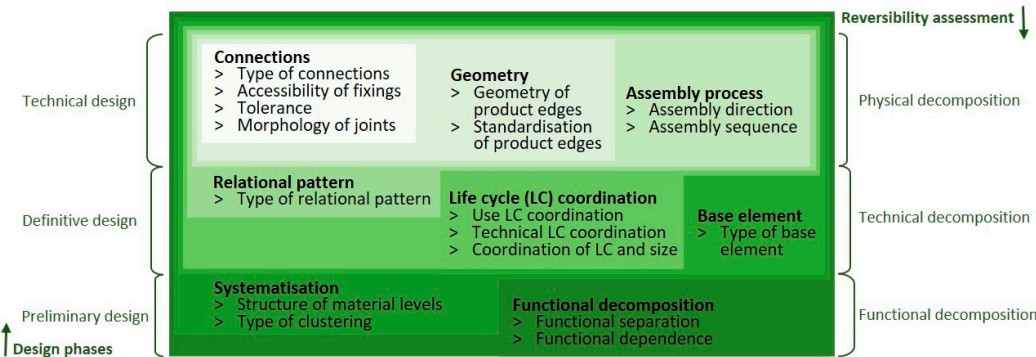

**Figure 2.** Reversible building design protocol for technical aspects of reversibility (own adaption of [33]), in which the type of connection is the first indicator at the core when applying the design protocol to assess the reversibility of a building.

The next section describes the applied assessment methods and the two case studies which the linked assessment method was applied. Section 3 presents the results of both case studies. Section 4 concludes this paper by discussing the conclusions based on the assessment results and the added value and limitations of the method applied.

## 2. Materials and Methods

### 2.1. Methodology

The methodology consists of three steps:

1. An LCA method based on EN 15804 and EN 15978 made specific for the Belgian construction sector, expressing the global warming potential (GWP) in kg $CO_2$ eq. and aggregated environmental impacts in external environmental costs [34–36];
2. An assessment of the detachability based on the type of connection as described within the design for disassembly framework by Durmisevic [37] and further developed by Van Vliet [38,39] and implemented in the BCI [10];
3. The mathematical linking of LCA to detachability assessment.

The next subsections explain in more detail the three steps of the methodology.

### 2.1.1. LCA Method

The LCA calculations are based on TOTEM (Tool to Optimise the Environmental impact of Materials) [40]—a free-access web tool launched in February 2018 by the three Belgian regional authorities, which mainly uses generic ecoinvent life cycle inventory data, and default transport and EOL scenarios as prescribed in the Belgian national supplement to the EN 15804 [41] over a building lifespan of 60 years.

For this study, version 2020 of the method behind TOTEM [35] is used as a methodological LCA framework based on EN 15804+A1:2013 [42] and EN 15978:2011 [24]. In addition to the environmental impact categories included in the EN 15804+A1, other impact categories included in the first version of PEF [43] were also considered to cover a broader environmental perspective than the seven impact categories in the +A1 version of the EN 15804. A single score can be calculated by means of an external environmental costing method. This means that each characterized individual indicator is multiplied with their corresponding monetization factor (see Table 1), resulting in an environmental cost indicator which subsequently allows aggregation of all monetized indicators to come to a single score expressed in euros. By applying this method, the EEC (external environmental costs) is taken as one of the two LCA indicators in the linking exploration. The EEC is expressed in the unit EUR/functional unit, i.e., $m^2$ wall in case of external walls. By taking

the EEC based on 17 impact categories, burden-shifting is minimized. By calculating a single score, however, the level of uncertainty regarding the results increases. Therefore, the midpoint indicator GWP (in the unit kg $CO_2$ eq.) is also considered in this exploration. The GWP life cycle impact assessment (LCIA) model is based on the most up-to-date and scientifically-robust consensus-based model by the IPCC [44]. Furthermore, this study uses the same generic life cycle inventory datasets based on ecoinvent 3.3 [45] and scenarios as used in TOTEM tool version 2.2.

**Table 1.** Overview of the applied environmental impact categories and the corresponding units, LCIA models and monetization factor [46].

| # | Impact Category | Unit | LCIA Model | Monetary Value [€/Unit] |
|---|---|---|---|---|
| 1 | Global warming | kg $CO_2$ eq. | EN 15804+A1 [42] (as used in CML version October 2012) | 0.05 |
| 2 | Ozone depletion | kg CFC 11 eq. | EN 15804+A1 [42] (as used in CML version October 2012) | 49.10 |
| 3 | Acidification for soil and water | kg $SO_2$ eq. | EN 15804+A1 [42] (as used in CML version October 2012) | 0.43 |
| 4 | Eutrophication | kg $(PO_4)_3$-eq. | EN 15804+A1 [42] (as used in CML version October 2012) | 20 |
| 5 | Photochemical ozone creation | kg Ethene eq. | EN 15804+A1 [42] (as used in CML version October 2012) | 0.48 |
| 6 | Depletion of abiotic resources—elements | kg Sb eq. | EN 15804+A1 [42] (as used in CML version October 2012) | 1.56 |
| 7 | Depletion of abiotic resources—fossil fuels | MJ, net calorific value | EN 15804+A1 [42] (as used in CML version October 2012) | 0 |
| 8a | Human toxicity—cancer effects | CTUh | Rosenbaum et al., 2008 [47] (as used in USEtox) | 665,109 |
| 8b | Human toxicity—non-cancer effects | CTUh | Rosenbaum et al., 2008 [47] (as used in USEtox) | 144,081 |
| 9 | Particulate matter | kg PM2.5 eq. | Rabl & Spandaro, 2004 [48] (RiskPoll) | 34 |
| 10 | Ionising radiation-human health effects | kg U235 eq. | Frischknecht et al., 2000 [49] (as used in ReCiPe midpoint) | $9.7 \times 10^{-4}$ |
| 11 | Ecotoxicity: freshwater | CTUe | Rosenbaum et al., 2008 [47] (as used in USEtox) | $3.7 \times 10^{-5}$ |
| 12 | Water resource depletion | $m^3$ water eq. | Frischknecht et al., 2008 [49] (as used in Swiss Ecoscarcity 2006) | 0.067 |
| 13a | Land use: occupation, soil organic matter | kg C deficit | Milà i Canals et al., 2007 [50] (Soil Organic Matter) | $1.4 \times 10^{-6}$ |
| 13b | Land use: occupation, biodiversity-all * | PDF·$m^2$yr | Köllner, 2000 [51] (as used in Eco-Indicator 99) | * |
| 13m1 | -urban | $m^2$yr | Köllner, 2000 [51]; characterisation factors set on (−)1 | 0.30 |
| 13m2 | -agricultural | $m^2$yr | Köllner, 2000 [51]; characterisation factors set on (−)1 | 0.006 |

**Table 1.** *Cont.*

| # | Impact Category | Unit | LCIA Model | Monetary Value [€/Unit] |
|---|---|---|---|---|
| 13m3 | -forest | $m^2$yr | Köllner, 2000 [51]; characterisation factors set on (−)1 | $2.2 \times 10^{-4}$ |
| 14a | Land use: transformation, soil organic matter | kg C deficit | Milà i Canals et al., 2007 [50] (Soil Organic Matter) | $1.4 \times 10^{-6}$ |
| 14b | Land use: transformation, biodiversity-all ** | PDF·$m^2$ | Köllner, 2000 [51] (as used in Eco-Indicator 99) | ** |
| 14m1 | -urban | $m^2$ | not available | not available |
| 14m2 | -agricultural | $m^2$ | not available | not available |
| 14m3 | -forest | $m^2$ | not available | not available |
| 14m4 | -tropical forest | $m^2$ | Köllner, 2000 [51]; characterisation factors set on (−)1 | 27 |

* The monetization factor for the indicator "land use occupation: biodiversity" is split up into three different sub-flows due to a lack of reliable monetary data for all the flows in one indicator. ** The monetization factor for the indicator "land use transformation: biodiversity" is not available for the sub-flows transformation from urban land, agricultural land and forest due to a lack of reliable monetary data. Geographically, the loss of tropical rainforest is not applicable within Europe, but still relevant. Therefore the monetary value for the region "rest of the world" is used as the default for the region Europe.

In addition, the modularity approach (see Figure 3), in line with EN 15804 and EN 15978, is applied. Each module covers a specific life cycle stage–e.g., information module A1 corresponds with the supply of raw materials, and A2 with the transportation of the raw materials to the factory. This study considers modules A1–A5, B2, B4, and C1–C4, with the exception of module B6 (indicated in Figure 3). By excluding module B6, this study focuses on the embodied environmental impact of the building solutions and streamlines the LCA on the aspect of operational energy use. The exclusion of module B6 is appropriate, as the variants assessed per the case study are designed with the same thermal resistance (in the same climatic region) and, therefore, would have the same environmental impact due to operational energy use.

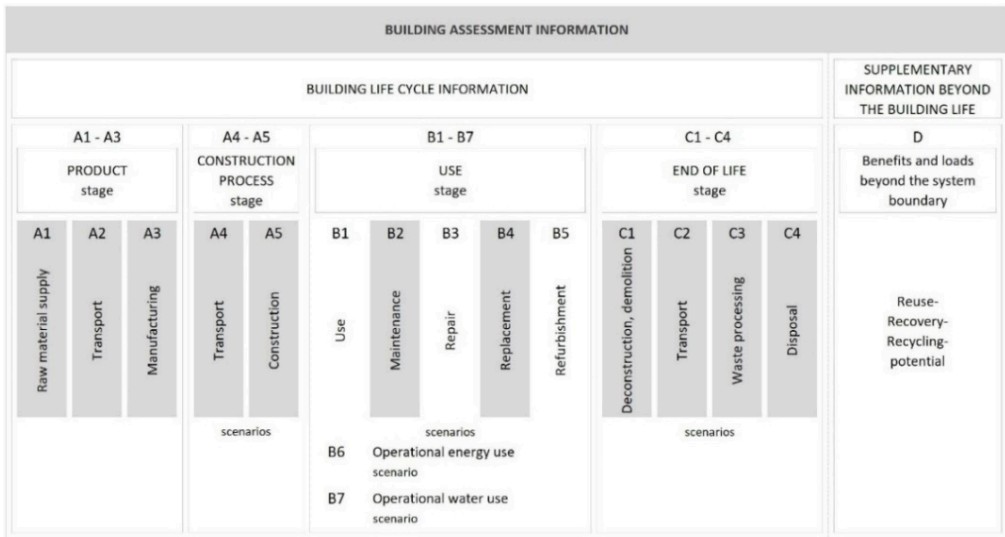

**Figure 3.** Overview of the life cycle stages and system boundaries in line with EN 15804 [23] and EN 15978 [24]. This study considers the information modules in the grey-colored boxes.

The life cycle inventory data on the amounts of material per building solutions used in the production stage is also used to determine the mass needed to represent the material input flow of the MFA. The output flow—i.e., construction waste at the EOL of the building—is not considered in this study, as in a linear life cycle, this would equal the input flow due to the mass balance and would result in 0. Thus, the mass of material use considered in the assessments of this study could also be interpreted as the amount of waste leaving the system at the EOL.

2.1.2. Detachability Assessment Based on Type of Connection

The detachability of a building solution is the degree to which objects can be disassembled without compromising the function of the object or surrounding objects in order to preserve the existing value [39]. Detachability is one of the key principles within the design strategy DfD [25]. DfD is a common ground that can be found in readings on circular building design [11,12,17,25,33,37,52,53]. Van Vliet et al. [39] developed a methodology based on the adaptation of Durmisevic [37] to measure detachability based on four aspects: type of connection (TC), accessibility of the connection (AC), interpenetrations (IP), and geometry of building solution edges (GE). Each aspect is scored from 0.1 to 1.0 based on qualitative conditions that are specified per aspect, with 0.1 corresponding to the lowest rating and 1.0 with the highest. Then, the detachability index (DI) of a building solution is calculated based on the harmonic mean of the four aspects—i.e., by dividing the number 4 (as there are four aspects) with the sum of the inverses of each score per aspect (see Equation (1)). By calculating the harmonic mean, low-scoring aspects will have a bigger influence on the detachability index of a building solution than when the average of the four aspects is calculated. Van Vliet et al. have developed their detachability index method specifically on the product level.

$$DI_{product,n} = \frac{4}{TC_n{}^{-1} + AC_n{}^{-1} + IP_n{}^{-1} + GE_n{}^{-1}} \tag{1}$$

To calculate a DI on the building level, they have provided an example of applying a weighting method based on the EEC of each product and the total EEC on the building level (see Equation (2)). However, without providing actual weighting factors that could be used and with the remark that further research should show whether a weighting method is necessary.

$$DI_{building} = \frac{EEC_{building}}{\sum EEC_{product,n} * DI_{product,n}} \tag{2}$$

As explained in the introduction (Section 1.4), this study considers only the detachability assessment based on the type of connection—hereafter referred to as Connection Index (CI). To assess the CI, the assessment criteria and scores as defined by Van Vliet et al. [39] have been applied and are presented in Table 2. A CI was determined for each material within the assessed case studies based on how each material is connected to the underlying or adjacent material, taking into consideration the prevention of double counting connections—e.g., in the case of paint on gypsum plaster boards that are screwed to a timber support structure, the paint is scored a 0.10, and the plasterboard is scored a 0.80.

**Table 2.** Assessment criteria and scores of the Connection Index (CI) [39].

| Type of Connection | | Score |
|---|---|---|
| Dry connection | Loose (no fixings) Click connection Velcro connection Magnetic connection | 1.00 |

**Table 2.** *Cont.*

| Type of Connection | | Score |
|---|---|---|
| Connection with added elements | Bolt and nut connection<br>Spring connection<br>Corner connection<br>Screw connection<br>Connection with added elements (e.g., a façade suspension system) | 0.80 |
| Direct integral connection | Pin connection (e.g., staples)<br>Nail connection | 0.60 |
| Soft chemical connection | Sealant connection<br>Foam connection | 0.20 |
| Hard chemical connection | Adhesive bond<br>Cast bond<br>Weld joint<br>Cement bond<br>Chemical anchors<br>Hard chemical bond | 0.10 |

### 2.1.3. Linking LCA Results to Detachability Assessment Scores

To support the assessment, the following four hierarchic levels of analysis have been distinguished: building, building elements, building components and building materials. Each level is the sum of parts of the underlying level (as illustrated in Figure 4).

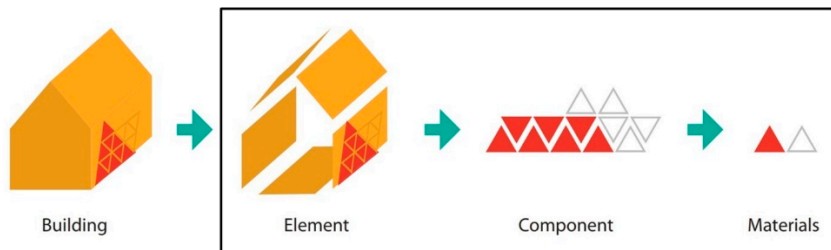

**Figure 4.** The hierarchical levels of analysis [36]. This study only considers the levels in the black rectangle.

The above-explained levels make it easier to comprehend a building assessment due to the possibility of a hierarchical decomposition of a building [54]. For a building LCA, the environmental impact (EI) of the materials is determined first, and the EI of the higher levels is based on the EI of the materials and the number of materials in the building, taking the hierarchical decomposition into consideration. In the case of a circularity assessment of a building, the same hierarchical decomposition can also be applied to analyze the detachability by assessing the CI of each material within a component first, followed by a (weighted) aggregation of the CI of each component to a CI on the element level.

As mentioned in the previous subsection, Van Vliet et al. [39] do not yet prescribe a method to calculate a DI on the building level and mention that further research on (the necessity of) a possible weighting method is needed. Regarding the principle of reversible connections, the ISO 20887:2020 standard [17] provides the possibility to measure that principle by determining the total percentage of connection types that can be reversed for material recovery based on a "yes or no" assessment. While for MFA-based circularity assessments, it is clear how to determine a score on the building level, there are no clear, consistent guidelines for non-MFA-based circularity indicators on how to aggregate and/or weigh circularity scores on a material level so as to achieve a higher building solution level of building solution. Therefore, this study has explored four possibilities of linking the potential aggregated EI (in the unit EUR/$m^2$), the GWP (in the unit kg $CO_2$ eq./$m^2$) or

mass input flows (M in the unit $kg/m^2$) to the unitless CI, and has furthermore explored how this can be used to calculate a weighted aggregated score on the element level.

Four possibilities of linking the EI, GWP or M to the CI are explored to investigate a simple objective method in which the CI is used as a factor, taking the above-described hierarchical decomposition in a circularity assessment into consideration. The first linking option is based on a simple multiplication of the indicators on the material level. In the other three options, the level of components is included, as the type of connection on the material level can influence the detachability of a component. By applying the four linking options, the relevance of the way of linking when it comes to drawing conclusions can be explored too. The four linking options are next explained in more detail:

1. The first option of linking is done by the multiplication (1-CI) of each material with the EI, GWP or M of each material. The CI is subtracted from one, as the higher the CI, the better. While in the case of the EI, GWP or M, the lower the amount, the better. The subtraction is therefore needed to align the variables. Thus, the following equation is used to calculate the linked EI (EI′) of a building element consisting of n materials:

$$EI'_{\text{building element}} = \sum EI_{\text{material, n}} \times \left(1 - CI_{\text{material, n}}\right) \tag{3}$$

To calculate the linked GWP (GWP′) or M (M′) of a building element, the same equation is used, but by replacing EI in the equation with GWP or M, respectively; this remark also applies to the other three equations.

2. In the second explored option, an average (AVG) CI per component is calculated first before subtracting it from 1 and subsequently multiplying it with the EI, GWP or M of each material. The average component CI is calculated based on the CIs of the materials within each component. The second equation is:

$$EI'_{\text{building eelement}} = \sum EI_{\text{material, n}} \times \left(1 - CI_{\text{componentAVG, m}}\right)$$

$$\text{where} : CI_{\text{componentAVG}} = \frac{\sum CI_{\text{material,n}}}{n_{\text{mateials}}} \tag{4}$$

3. As a variant on the previous option, in the third option, the smallest (i.e., worst scoring; MIN) CI of a material within a component is taken instead of the AVG of a component:

$$EI'_{\text{building element}} = \sum EI_{\text{material, n}} \times \left(1 - CI_{\text{componentMIN, m}}\right)$$

$$\text{where} : CTD_{componentMIN} = MIN\left(CI_{product,n}\right) \tag{5}$$

4. In the final possibility, the harmonic mean (HM) of CIs per component is applied:

$$EI'_{\text{building element}} = \sum EI_{\text{material, n}} \times \left(1 - CI_{\text{componentHM, m}}\right)$$

$$\text{where} : CI_{\text{componentHM}} = \frac{n_{\text{materials}}}{\sum \frac{1}{CI_{\text{material,n}}}} \tag{6}$$

### 2.2. Case Study Descriptions

For this study, two case studies are taken from two different projects in which external wall variants have been designed and LCA have been conducted. The case studies cover both new construction and renovation and were selected on the basis of the widespread construction methods applied, in one case, brickwork with external cladding, and in the other case, a concrete structure with infill elements. Both case studies are located in Belgium: Belgium has a temperate maritime climate characterized by moderate temperatures, prevailing southerly to westerly winds, abundant cloud cover and frequent precipitation. Summers are relatively cool and humid, and winters are relatively mild and rainy (climatic

zone Cfb according to Köppen) [55]. The first case study concerns a primary school designed as a prototype for the Circular School of the Future (rendering included as Figure 5) to be built in Brasschaat, Antwerp. The CI analysis was done after the project was already finished. The second case study is a renovation of a student housing building in Brussels, of which both LCA and CI were performed during the project. The two next subsections present the assessed external wall variants in more detail.

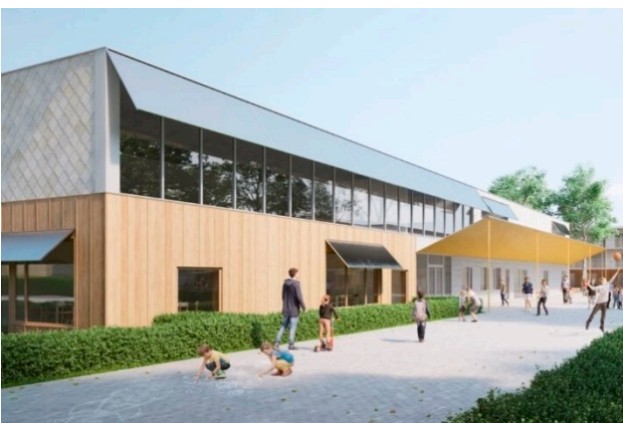

**Figure 5.** Case study 1: Primary school De Kaart in Brasschaat, province of Antwerp (Belgium). Source: group Van Roey, Compagnie O and Cluster [56].

### 2.2.1. Case Study 1: New Built Primary School

In the first case study, a business-as-usual (BAU) external wall was designed first, followed by four alternative designs. The BAU configuration consists of an inner brick wall, thermal insulation on the outside and a brick, wooden or slate external finish. This configuration is typical for the Belgian building stock and is characterized by its simplicity and price. Moreover, the wall configuration provides sufficient inertia to the building. However, this composition mainly consists of non-renewable materials and the connections typically used—such as cement mortar—are irreversible. In the current practice, when building this BAU wall design, future adaptations—such as the addition or enlargement of existing windows—require the demolition of existing building materials and creates downcycled demolition waste.

The four alternative wall variants included in this paper are designed as more circular solutions by applying renewable materials and/or reversible connections. The LCA was used to check whether the alternatives are also more sustainable from an EI point of view. To keep the same look and feel on the outside and inside of all variants, the external and internal finish is kept the same. Hence, the main difference between the four alternative solutions focuses on the load-bearing part of the wall configuration (see also Table 3 and Figure 6).

In the first wall variant, only the cement mortar of the BAU composition is replaced by a lime mortar. Typically, lime mortar is less strong compared to cement mortar, but it improves the reversibility of the connection. To further improve the reversibility of the wall design, an alternative for the bricks is considered in the second and third wall variants—i.e., hempcrete blocks and a wooden building block system, respectively. The hempcrete blocks are renewable and provide inherent insulation characteristics. The wooden building blocks are filled with loose insulation flakes, but an extra insulation board is added on the outside to reach equal thermal properties. In the last alternative design, timber prefab load-bearing components filled with insulation flakes are used. As is the case for the third alternative, the use of wood and loose insulation materials offers additional advantages for creating a more reversible solution. It also allows the use of a more reversible connection type (i.e., bolts) compared to the previously described variants.

**Table 3.** The main components of the four external wall variants included in case study 1 are newly built primary schools.

| Wall Variant | External Finish | Insulation | Load-Bearing Structure | Internal Finish |
|---|---|---|---|---|
| **Variant 1.1** lime mortar | Varnish Softwood cladding | PUR board | Clay brick with lime mortar | Gypsum plaster Paint |
| **Variant 1.2** hempcrete | Varnish Softwood cladding | Hempcrete blocks with adhesive mortar | | Lime plaster Paint |
| **Variant 1.3** wooden blocks | Varnish Softwood cladding | Wooden building blocks filled w/cellulose flakes PUR board | | Gypsum plaster board Paint |
| **Variant 1.4** prefab timber | Varnish Softwood cladding | Prefab timber components filled with stone wool blankets | | Gypsum plaster board Paint |

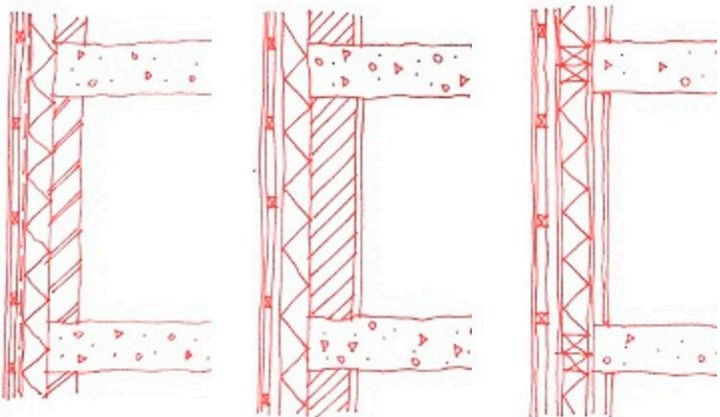

**Figure 6.** Schematic sections of the wall variants included in case study 1—the newly built primary school. On the left, variant 1.1 (clay masonry with lime mortar); in the middle variants 1.2 and 1.3 (alternative masonry with hempcrete and wooden blocks, respectively); and on the right variant 1.4 (prefab timber components). Source: Stijn Elsen, VUB Architectural Engineering [56].

2.2.2. Case Study 2: Renovation of Student Housing Building

The second case study is the design of different renovation strategies for student housing buildings by architect Willy Van Der Meeren at VUB—also known as the WVDM Living Lab. The WVDM student housing buildings (see Figure 7) were designed between 1971 and 1973, and have a big cultural historic value for VUB and the city of Brussels. With the WVDM Living Lab, VUB wants to realise an economically feasible and future-oriented renovation, in which sustainability and energy performance requirements are met, while at the same time preserving its cultural heritage.

The research consortium defined four research strategies, based upon which they made the material selection for the renovation of the external walls. The research strategies were based on a combination of the four evaluation pillars as defined by VUB: heritage, economy, energy and sustainability. The first strategy consists of a combination of heritage and economy, in which the heritage is preserved at a maximum, new material input is included at a minimum, and the refurbishment realizes basic comfort for the users of the student housing. The second strategy combines the pillars energy and heritage, and aims at energy efficiency, specific insulation measures and basic comfort. The third strategy focuses first on sustainability and secondly on energy, and includes detachable compositions with environmentally friendly materials, high energy efficiency and adaptive comfort. The last strategy also combines the same two pillars as the third one, but places energy above sustainability and includes aspects of a (passive) deep energy renovation, minimal environmental impact and high comfort.

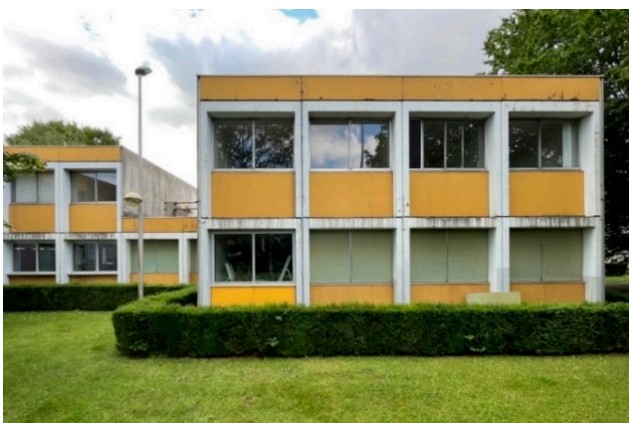

**Figure 7.** Case study 2: WVDM Living Lab, renovation student housing buildings Vrije Universiteit Brussel, Brussels (Belgium). Source: MAKER Architecten [57].

The four presented external wall compositions were designed and assessed as façade modules of 8.35 m², to be integrated into the existing concrete structure and consisting of columns, beams and floors (see Figure 8). For this paper, only the newly added materials have been considered, and the results have been calculated for 1 m² wall. The existing structural components have been left out of scope, as the weight of the concrete structure outweighs the new materials and is the same in each strategy. Additionally, because of the recycle content method considered in the LCA method, the existing components are regarded as burden-free. The EI of the demolition of existing components in strategies 2, 3 and 4, which is limited, is also excluded from the results in this paper, so the focus lies on the materials included in the compositions and their connection type. Each strategy also included a specific proposal for a new building-related energy system. For the purpose of this paper, the energy system is left out of scope too, with the exception of the building integrated photovoltaic (BIPV) panels in strategy 4 that are used as the external finish. The main components in the four strategies—i.e., wall variants—are listed in Table 4.

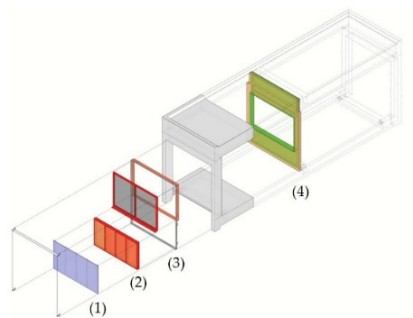

**Figure 8.** 3D model of the façade module of case study 2 student housing renovation, consisting of sub-modules: (1) external finish, (2) insulation, (3) external window and (4) internal finish. Source: MAKER Architecten [57].

**Table 4.** The main components of the four external wall variants included in case study 2 student housing renovation. The existing components are indicated with *. The numbers between () in the top row correspond with the numbers in Figure 8.

| Wall Variant | External Finish (1) | Insulation (2) | External Window (3) | Internal Finish (4) |
|---|---|---|---|---|
| **Variant 2.1** Heritage + Economy | Concrete panel * | Polystyrene (above + under window) * | Aluminium frame * | Hollow wood fibreboard * |
| | | | Single glazing * | Hermetic, aluminium frame with single glazing |

**Table 4.** *Cont.*

| Wall Variant | External Finish (1) | Insulation (2) | External Window (3) | Internal Finish (4) |
|---|---|---|---|---|
| **Variant 2.2** Energy + Heritage | 3D printed panels | Stone wool in timber frame | Steel frame | Detachable steel frame with resol and stone wool filling and MDF finish |
| | Concrete panel * | | Double glazing | |
| **Variant 2.3** Sustainability + Energy | Ceramic fibre-reinforced panels | Stone wool in timber frame | Hardwood frame | Detachable steel frame stone wool filling and MDF finish |
| | Concrete panel * | | Double glazing | |
| **Variant 2.4** Energy + Sustainability | BIPV panels | Hemp lime between timber laths | Hardwood frame | Lime plaster |
| | Concrete panel * | Stone wool in wood-aluminium curtain wall | Double glazing | |

## 3. Results

### 3.1. Unlinked Baseline Results

Tables 5 and 6 provide the unlinked assessment results for the four external wall variants of case study 1 and the four variants of case study 2, respectively. The materials of each wall variant are listed in the first column, followed by the component number in the second column—indicating which materials form a component within an element variant. The third column shows the aggregated environmental impact (EI) expressed as external environmental costs (€) per m$^2$ wall over the total life cycle of the material applied in the external wall variants. Although it is possible to provide the EI per life cycle stage with the modularity approach of the applied LCA methodology, only the EI aggregated over the total life cycle is provided, as the goal of this study is to explore the possibility and effect of linking quantitative LCA results to a simplified circularity rating. A more detailed assessment of the EI of each wall variant is not an objective of this paper. The fourth and fifth columns provide the global warming potential (GWP) in kg $CO_2$ eq. and mass input flows (M) in kilograms, respectively, both per m$^2$ wall. The EI, GWP and M figures in Tables 5 and 6 are not linked to the CI and serve as a baseline reference for the linking exploration. Per the wall variant, the total EI, total GWP and total M are also included in the tables. To conclude, the four last columns give the four CIs used in the linking exploration—i.e., the CI determined for each material (mat_CI), the average of material CIs within a component (AVR_CI), the worst material CI within a component (MIN_CI) and the harmonic mean of material CIs within a component (HM_CI).

**Table 5.** Baseline aggregated environmental impact (EI) results, global warming potential (GWP), mass input flows (M), and four options of connection indexes (CI; mat_CI = CI on the material level, AVR_CI = average component CI, MIN_CI = the minimal CI within a component, HM_CI = harmonic mean component CI) of all external wall variants of case study 1, in absolute values.

| Wall Variant Material | Component # | EI [€/m$^2$] | GWP [kg. $CO_2$ eq./m$^2$] | M [€/m$^2$] | mat_CI [-] | AVR_CI [-] | MIN_CI [-] | HM_CI [-] |
|---|---|---|---|---|---|---|---|---|
| **VAR1.1** lime mortar masonry | | | | | | | | |
| Varnish | 1 | 2.65 | 17.25 | 0.3 | 0.10 | 0.10 | 0.10 | 0.10 |
| Softwood cladding, screwed | 2 | 3.27 | 8.88 | 13.2 | 0.80 | 0.80 | 0.80 | 0.80 |
| Timber substructure, screwed | 2 | 0.29 | 0.79 | 2.7 | 0.80 | 0.80 | 0.80 | 0.80 |
| PUR insulation board | 3 | 4.06 | 33.90 | 3.8 | 0.80 | 0.80 | 0.60 | 0.77 |
| Nylon fixings insulation board | 3 | 0.05 | 0.60 | 0.0 | 1.00 | 0.80 | 0.60 | 0.77 |
| Steel fixings insulation board | 3 | 0.02 | 0.08 | 0.0 | 0.60 | 0.80 | 0.60 | 0.77 |
| Clay brick | 4 | 2.67 | 31.66 | 124.6 | 0.10 | 0.10 | 0.10 | 0.10 |
| Lime mortar | 4 | 1.32 | 15.24 | 24.5 | 0.10 | 0.10 | 0.10 | 0.10 |
| Gypsum plaster | 5 | 0.40 | 3.66 | 10.8 | 0.10 | 0.10 | 0.10 | 0.10 |
| Softwood cladding, screwed | 6 | 3.74 | 17.98 | 0.3 | 0.10 | 0.10 | 0.10 | 0.10 |

**Table 5.** *Cont.*

| Wall Variant Material | Component # | EI [€/m²] | GWP [kg. CO₂ eq./m²] | M [€/m²] | mat_CI [-] | AVR_CI [-] | MIN_CI [-] | HM_CI [-] |
|---|---|---|---|---|---|---|---|---|
| | **VAR1.1 TOTAL** | **18.46** | **130.04** | **180.2** | | | | |
| **VAR1.2** hempcrete blocks | | | | | | | | |
| Varnish | 1 | 2.65 | 17.25 | 0.3 | 0.10 | 0.10 | 0.10 | 0.10 |
| Softwood cladding, screwed | 2 | 3.27 | 8.88 | 13.2 | 0.80 | 0.80 | 0.80 | 0.80 |
| Timber substructure, screwed | 2 | 0.29 | 0.79 | 2.7 | 0.80 | 0.80 | 0.80 | 0.80 |
| Hempcrete blocks | 3 | 3.60 | 38.31 | 46.9 | 0.10 | 0.10 | 0.10 | 0.10 |
| Adhesive mortar | 3 | 0.07 | 0.81 | 3.8 | 0.10 | 0.10 | 0.10 | 0.10 |
| Lime plaster | 4 | 1.65 | 17.56 | 16.2 | 0.10 | 0.10 | 0.10 | 0.10 |
| Paint | 5 | 3.74 | 17.98 | 0.3 | 0.10 | 0.10 | 0.10 | 0.10 |
| | **VAR1.2 TOTAL** | **15.26** | **101.58** | **83.3** | | | | |
| **VAR1.3** wooden blocks | | | | | | | | |
| Varnish | 1 | 2.65 | 17.25 | 0.3 | 0.10 | 0.10 | 0.10 | 0.10 |
| Softwood cladding, screwed | 2 | 3.27 | 8.88 | 13.2 | 0.80 | 0.80 | 0.80 | 0.80 |
| Timber substructure, screwed | 2 | 0.29 | 0.79 | 2.7 | 0.80 | 0.80 | 0.80 | 0.80 |
| PUR insulation board, screwed | 3 | 1.04 | 5.18 | 2.6 | 0.80 | 0.80 | 0.80 | 0.80 |
| Wooden building blocks | 4 | 5.62 | 22.36 | 44.8 | 0.80 | 0.90 | 0.80 | 0.89 |
| Cellulose insulation flakes, loose | 4 | 1.04 | 5.18 | 9.3 | 1.00 | 0.90 | 0.80 | 0.89 |
| Timber substructure, screwed | 5 | 0.30 | 0.80 | 2.7 | 0.80 | 0.80 | 0.80 | 0.80 |
| Gypsum plaster board, screwed | 6 | 1.22 | 8.66 | 9.1 | 0.80 | 0.45 | 0.10 | 0.18 |
| Gypsum jointing compound | 6 | 0.04 | 0.25 | 0.2 | 0.10 | 0.45 | 0.10 | 0.18 |
| Paint | 7 | 3.74 | 17.98 | 0.3 | 0.10 | 0.10 | 0.10 | 0.10 |
| | **VAR1.3 TOTAL** | **19.20** | **87.35** | **84.9** | | | | |
| **VAR1.4** prefab wooden components | | | | | | | | |
| Varnish | 1 | 2.65 | 17.25 | 0.3 | 0.10 | 0.10 | 0.10 | 0.10 |
| Softwood cladding, screwed | 2 | 3.27 | 8.88 | 13.2 | 0.80 | 0.80 | 0.80 | 0.80 |
| Timber substructure, screwed | 2 | 0.29 | 0.79 | 2.7 | 0.80 | 0.80 | 0.80 | 0.80 |
| Bituminised fibreboard, prefab component | 3 | 8.41 | 31.54 | 5.9 | 0.80 | 0.85 | 0.80 | 0.84 |
| Timber frame, prefab component, screwed | 3 | 1.31 | 3.56 | 12.2 | 0.80 | 0.85 | 0.80 | 0.84 |
| Stone wool insulation, prefab component | 3 | 1.02 | 8.27 | 5.8 | 1.00 | 0.85 | 0.80 | 0.84 |
| OSB board of prefab component, screwed | 3 | 1.37 | 7.25 | 9.0 | 0.80 | 0.85 | 0.80 | 0.84 |
| Timber substructure, screwed | 4 | 0.30 | 0.80 | 2.7 | 0.80 | 0.80 | 0.80 | 0.80 |
| Gypsum plaster board, screwed | 5 | 1.22 | 8.66 | 9.1 | 0.80 | 0.45 | 0.10 | 0.18 |
| Gypsum jointing compound | 5 | 0.04 | 0.25 | 0.2 | 0.10 | 0.45 | 0.10 | 0.18 |
| Paint | 6 | 3.74 | 17.98 | 0.3 | 0.10 | 0.10 | 0.10 | 0.10 |
| | **VAR1.4 TOTAL** | **23.62** | **105.23** | **61.2** | | | | |

**Table 6.** Baseline aggregated environmental impact (EI) results, global warming potential (GWP), mass input flows (M), and four options of connection indexes (CI; mat_CI = CI on the material level, AVR_CI = average component CI, MIN_CI = the minimal CI within a component, HM_CI = harmonic mean component CI) of all external wall variants of case study 2, in absolute values.

| Wall Variant Material | Component # | EI [€/m$^2$] | GWP [kg. CO$_2$ eq./m$^2$] | M [€/m$^2$] | mat_CI [-] | AVR_CI [-] | MIN_CI [-] | HM_CI [-] |
|---|---|---|---|---|---|---|---|---|
| **VAR2.1 heritage + economy** | | | | | | | | |
| Existing concrete panel | 1 | 0.00 | 0.00 | 51.2 | 0.80 | 0.60 | 0.10 | 0.24 |
| Existing polystyrene strips | 1 | 0.00 | 0.00 | 0.2 | 1.00 | 0.60 | 0.10 | 0.24 |
| Existing PE vapour barrier, glued | 1 | 0.00 | 0.00 | 0.0 | 0.10 | 0.60 | 0.10 | 0.24 |
| Existing waterproof foam tape + sealant | 1 | 0.00 | 0.00 | 0.0 | 0.10 | 0.60 | 0.10 | 0.24 |
| Existing hollow wood fibre board | 1 | 0.00 | 0.00 | 5.1 | 0.80 | 0.60 | 0.10 | 0.24 |
| Existing aluminium sliding window | 1 | 0.00 | 0.00 | 8.7 | 0.80 | 0.60 | 0.10 | 0.24 |
| Aluminium frame | 2 | 14.20 | 88.94 | 7.2 | 0.80 | 0.93 | 0.80 | 0.92 |
| Single glazing | 2 | 5.02 | 33.23 | 1.1 | 1.00 | 0.93 | 0.80 | 0.92 |
| Demountable rubbers and tape | 2 | 0.55 | 4.31 | 9.1 | 1.00 | 0.93 | 0.80 | 0.92 |
| **VAR2.1 TOTAL** | | **19.77** | **126.48** | **82.6** | | | | |
| **VAR2.2 energy + heritage** | | | | | | | | |
| 3D printed panels | 1 | 5.60 | 18.10 | 5.8 | 1.00 | 0.90 | 0.80 | 0.89 |
| Vertical therm. modified wood laths | 1 | 0.10 | 0.27 | 0.2 | 0.80 | 0.90 | 0.80 | 0.89 |
| Existing concrete panel | 2 | 0.00 | 0.00 | 51.2 | 0.80 | 0.80 | 0.80 | 0.80 |
| Prefab steel structural frame | 3 | 1.41 | 8.44 | 4.0 | 1.00 | 0.55 | 0.10 | 0.18 |
| Waterproofing | 3 | 0.09 | 0.76 | 0.1 | 0.10 | 0.55 | 0.10 | 0.18 |
| Prefab hardwood timber frame | 4 | 0.02 | 0.05 | 1.3 | 0.80 | 0.45 | 0.10 | 0.18 |
| Air sealing tape | 4 | 0.00 | 0.02 | 0.0 | 0.10 | 0.45 | 0.10 | 0.18 |
| Timber frame | 5 | 0.09 | 0.30 | 0.7 | 0.60 | 0.57 | 0.10 | 0.24 |
| Stone wool insulation, timber frame | 5 | 0.17 | 1.40 | 1.1 | 1.00 | 0.57 | 0.10 | 0.24 |
| OSB board, internal finish | 5 | 0.36 | 2.17 | 2.9 | 0.80 | 0.57 | 0.10 | 0.24 |
| Woodfibre board, external finish | 5 | 0.31 | 1.30 | 1.5 | 0.80 | 0.57 | 0.10 | 0.24 |
| Waterproofing | 5 | 0.09 | 0.76 | 0.1 | 0.10 | 0.57 | 0.10 | 0.24 |
| Air sealing tape | 5 | 0.00 | 0.02 | 0.0 | 0.10 | 0.57 | 0.10 | 0.24 |
| Steel window frame with double glazing | 6 | 4.84 | 33.90 | 18.1 | 0.80 | 0.45 | 0.10 | 0.18 |
| Waterproofing | 6 | 0.07 | 0.56 | 0.1 | 0.10 | 0.45 | 0.10 | 0.18 |
| Steel T-shaped anchor | 7 | 0.06 | 0.36 | 0.2 | 0.80 | 0.80 | 0.80 | 0.80 |
| Sunblind, roller | 7 | 0.26 | 1.72 | 1.5 | 0.80 | 0.80 | 0.80 | 0.80 |
| Reusable modular interior wall system | 8 | 1.70 | 5.05 | 1.9 | 1.00 | 0.95 | 0.80 | 0.94 |
| Stone wool insulation, timber frame | 8 | 0.16 | 1.30 | 1.1 | 1.00 | 0.95 | 0.80 | 0.94 |
| Resol insulation | 8 | 0.50 | 7.03 | 1.1 | 1.00 | 0.95 | 0.80 | 0.94 |
| MDF board, internal finish | 8 | 0.07 | 2.14 | 5.0 | 0.80 | 0.95 | 0.80 | 0.94 |
| **VAR2.2 TOTAL** | | **15.89** | **85.64** | **97.9** | | | | |
| **VAR2.3 sustainability + energy** | | | | | | | | |
| Ceramic fibre-reinforced panels | 1 | 0.20 | 1.63 | 6.3 | 1.00 | 0.90 | 0.80 | 0.89 |
| Vertical therm. modified wood laths | 1 | 0.10 | 0.27 | 0.2 | 0.80 | 0.90 | 0.80 | 0.89 |
| Existing concrete panel | 2 | 0.00 | 0.00 | 51.2 | 0.80 | 0.80 | 0.80 | 0.80 |
| Prefab steel structural frame | 3 | 1.41 | 8.44 | 4.0 | 1.00 | 0.55 | 0.10 | 0.18 |
| Waterproofing | 3 | 0.09 | 0.76 | 0.1 | 0.10 | 0.55 | 0.10 | 0.18 |
| Prefab hardwood timber frame | 4 | 0.02 | 0.05 | 1.3 | 0.80 | 0.45 | 0.10 | 0.18 |
| Air sealing tape | 4 | 0.00 | 0.02 | 0.0 | 0.10 | 0.45 | 0.10 | 0.18 |
| Timber frame | 5 | 0.09 | 0.30 | 0.7 | 0.60 | 0.57 | 0.10 | 0.24 |
| Stone wool insulation, timber frame | 5 | 0.17 | 1.40 | 1.1 | 1.00 | 0.57 | 0.10 | 0.24 |
| OSB board, internal finish | 5 | 0.36 | 2.17 | 2.9 | 0.80 | 0.57 | 0.10 | 0.24 |
| Woodfibre board, external finish | 5 | 0.31 | 1.30 | 1.5 | 0.80 | 0.57 | 0.10 | 0.24 |

**Table 6.** *Cont.*

| Wall Variant Material | Component # | EI [€/m²] | GWP [kg. CO₂ eq./m²] | M [€/m²] | mat_CI [-] | AVR_CI [-] | MIN_CI [-] | HM_CI [-] |
|---|---|---|---|---|---|---|---|---|
| Waterproofing | 5 | 0.09 | 0.76 | 0.1 | 0.10 | 0.57 | 0.10 | 0.24 |
| Air sealing tape | 5 | 0.00 | 0.02 | 0.0 | 0.10 | 0.57 | 0.10 | 0.24 |
| Hardwood window frame with double glazing | 6 | 10.70 | 52.56 | 11.9 | 0.80 | 0.45 | 0.10 | 0.18 |
| Waterproofing | 6 | 0.07 | 0.56 | 0.1 | 0.10 | 0.45 | 0.10 | 0.18 |
| Steel T-shaped anchor | 7 | 0.06 | 0.36 | 0.2 | 0.80 | 0.80 | 0.80 | 0.80 |
| Sunblind, roller | 7 | 0.26 | 1.72 | 1.5 | 0.80 | 0.80 | 0.80 | 0.80 |
| Reusable modular interior wall system | 8 | 1.70 | 5.05 | 0.4 | 1.00 | 0.93 | 0.80 | 0.92 |
| Stone wool insulation, timber frame | 8 | 0.17 | 1.40 | 0.4 | 1.00 | 0.93 | 0.80 | 0.92 |
| Resol insulation | 8 | 0.35 | 2.14 | 1.0 | 0.80 | 0.93 | 0.80 | 0.92 |
| MDF board, internal finish | 8 | 0.00 | 0.00 | 0.0 | 0.00 | 0.00 | 0.00 | 0.00 |
| | **VAR2.3 TOTAL** | **16.16** | **80.89** | **85.0** | | | | |
| **VAR2.4** energy + sustainability | | | | | | | | |
| BIPV panels | 1 | 17.96 | 110.73 | 8.0 | 0.80 | 0.80 | 0.80 | 0.80 |
| Existing concrete panel | 2 | 0.00 | 0.00 | 51.2 | 0.80 | 0.80 | 0.80 | 0.80 |
| Timber laths, hemp lime | 3 | 0.34 | 0.94 | 2.9 | 0.60 | 0.50 | 0.10 | 0.23 |
| Hemp lime | 3 | 1.48 | 16.64 | 23.4 | 0.10 | 0.50 | 0.10 | 0.23 |
| Gypsum fibreboard | 3 | 0.73 | 4.70 | 9.4 | 0.80 | 0.50 | 0.10 | 0.23 |
| Wood-aluminium curtain wall profiles | 4 | 1.70 | 28.68 | 10.5 | 0.80 | 0.63 | 0.10 | 0.24 |
| Stone wool panels, curtain wall | 4 | 0.41 | 3.34 | 2.9 | 1.00 | 0.63 | 0.10 | 0.24 |
| Waterproofing | 4 | 0.09 | 0.76 | 0.1 | 0.10 | 0.63 | 0.10 | 0.24 |
| Hardwood window frame with double glazing | 5 | 12.43 | 60.70 | 13.5 | 0.80 | 0.45 | 0.10 | 0.18 |
| Waterproofing | 5 | 0.07 | 0.61 | 0.1 | 0.10 | 0.45 | 0.10 | 0.18 |
| Steel T-shaped anchor | 6 | 0.06 | 0.36 | 0.2 | 0.80 | 0.80 | 0.80 | 0.80 |
| Venetian blind, aluminium slats | 6 | 11.68 | 37.80 | 3.3 | 0.80 | 0.80 | 0.80 | 0.80 |
| Lime plaster, on hemp lime | 7 | 0.30 | 1.60 | 6.4 | 0.10 | 0.10 | 0.10 | 0.10 |
| | **VAR2.4 TOTAL** | **47.25** | **266.88** | **132.0** | | | | |

Based on Table 5 of case study 1, VAR1.2—the wall variant with the hempcrete blocks—is the variant with the lowest EI. VAR1.3—the variant with wooden blocks—is the variant with the lowest GWP. VAR1.4—with the prefab wooden components—is the variant with the lowest M; however, it is also the variant with the highest EI and second-highest GWP. When looking at the results of case study 2 in Table 6, VAR2.2—the strategy in which energy and heritage are the leading evaluation pillars—is the variant with the lowest EI. The variant with the lowest GWP is VAR2.3, which consists of a combination of the evaluation pillars of sustainability and energy. VAR2.1—which combines the evaluation pillars heritage and economy—is the option with the lowest M.

### 3.2. Applying the Linking Method

Table 7 shows the assessment results when the EI, GWP or M are linked to the CI, in absolute values as well as in terms of the relative difference to the baseline. The absolute values grouped per external wall variant are also presented in Figure 9. When looking at the linked results of case study 1, a shift of which wall variant would be more favourable can be seen: the two variants with higher Cis—in other words, with more reversible connection types (VAR1.3 and VAR1.4)—have higher relative differences with their respective baselines:

- between 50% and 62% regarding the EI,
- between 45% and 58% regarding the GWP,

- between 69% and 82% regarding the M,

**Table 7.** Linked EI, GWP and M results of all external wall variants for all four linking options, in absolute values and in relative difference to the baseline.

| | | VAR1.1 | VAR1.2 | VAR1.3 | VAR1.4 | VAR2.1 | VAR2.2 | VAR2.3 | VAR2.4 |
|---|---|---|---|---|---|---|---|---|---|
| **Linked with HM_CI** | M*(1-HM_CI) | −18% | −23% | −78% | −72% | −39% | −63% | −64% | −49% |
| | | 148.5 | 63.9 | 18.3 | 17 | 50.8 | 36 | 30.7 | 67.2 |
| | GWP*(1-HM_CI) | −33% | −17% | −48% | −53% | −92% | −49% | −29% | −54% |
| | | 87.24 | 84.66 | 45.21 | 49.1 | 9.73 | 43.81 | 57.07 | 123.5 |
| | EI*(1-HM_CI) | −38% | −26% | −56% | −60% | −92% | −57% | −31% | −57% |
| | | 11.38 | 11.25 | 8.5 | 9.47 | 1.52 | 6.89 | 11.14 | 20.11 |
| **Linked with MIN_CI** | M*(1-MIN_CI) | −17% | −23% | −72% | −69% | −25% | −60% | −62% | −44% |
| | | 149.1 | 63.9 | 23.8 | 19.1 | 62.2 | 39.4 | 32.4 | 73.3 |
| | GWP*(1-MIN_CI) | −28% | −17% | −45% | −51% | −80% | −44% | −21% | −57% |
| | | 92.98 | 84.66 | 48.35 | 51.92 | 25.3 | 48.22 | 63.63 | 113.8 |
| | EI*(1-MIN_CI) | −35% | −26% | −52% | −57% | −80% | −54% | −23% | −62% |
| | | 12.06 | 11.25 | 9.19 | 10.07 | 3.95 | 7.25 | 12.49 | 18.15 |
| **Linked with AVG_CI** | M*(1-AVG_CI) | −18% | −23% | −82% | −77% | −67% | −73% | −73% | −64% |
| | | 148.3 | 63.9 | 15.2 | 14.2 | 27.3 | 26.4 | 22.7 | 47.0 |
| | GWP*(1-AVG_CI) | −34% | −17% | −51% | −56% | −93% | −69% | −54% | −75% |
| | | 86.07 | 84.66 | 42.49 | 46.29 | 8.43 | 26.9 | 36.96 | 65.96 |
| | EI*(1-AVG_CI) | −39% | −26% | −58% | −62% | −93% | −75% | −55% | −75% |
| | | 11.24 | 11.25 | 8.08 | 9.03 | 1.32 | 4.01 | 7.26 | 11.58 |
| **Linked with mat_CI** | M*(1-CI) | −18% | −23% | −82% | −81% | −82% | −83% | −82% | −64% |
| | | 148.3 | 63.9 | 15.6 | 11.6 | 14.5 | 17 | 15 | 47.5 |
| | GWP*(1-CI) | −34% | −17% | −53% | −58% | −86% | −88% | −83% | −75% |
| | | 85.96 | 84.66 | 41.27 | 44.23 | 17.79 | 10.41 | 14.14 | 66.62 |
| | EI*(1-CI) | −39% | −26% | −58% | −62% | −86% | −91% | −83% | −77% |
| | | 11.23 | 11.25 | 8.13 | 9.02 | 2.84 | 1.46 | 2.69 | 10.8 |
| | | **VAR1.1** | **VAR1.2** | **VAR1.3** | **VAR1.4** | **VAR2.1** | **VAR2.2** | **VAR2.3** | **VAR2.4** |

Compared to the glued wall alternatives (VAR1.1 and VAR1.2, i.e., only between 26% and 39% on the EI, between 17% and 34% on the GWP and between 17% and 23% on the M). A comparable shift does not occur in the linked results of case study 2: the variants with the lowest baseline EI, GWP or M stay the variants with the lowest linked EI′, GWP′ or M′.

In the linked results of case study 2, a bigger variation between the four linking options is noticeable per wall variant, which is not the case with the linked results of case study 1. For instance, the relative difference of the linked EI to the baseline of VAR2.3 drops from 83% when the EI is linked with the CI on the material level to 55% when the EI is linked on the average component CI, and even to only 23% when linked with the minimum CI within a component. Such a variation in relative difference to the linked M is also visible when looking at the figures of VAR2.1: −82% when linked with the mat_CI and only −25% when linked with the MIN_CI. When looking at the linked GWP results of VAR2.1, the variation in relative difference shows a similar tendency: −93% when linked with the average component CI and only −25% when linked with the MIN_CI.

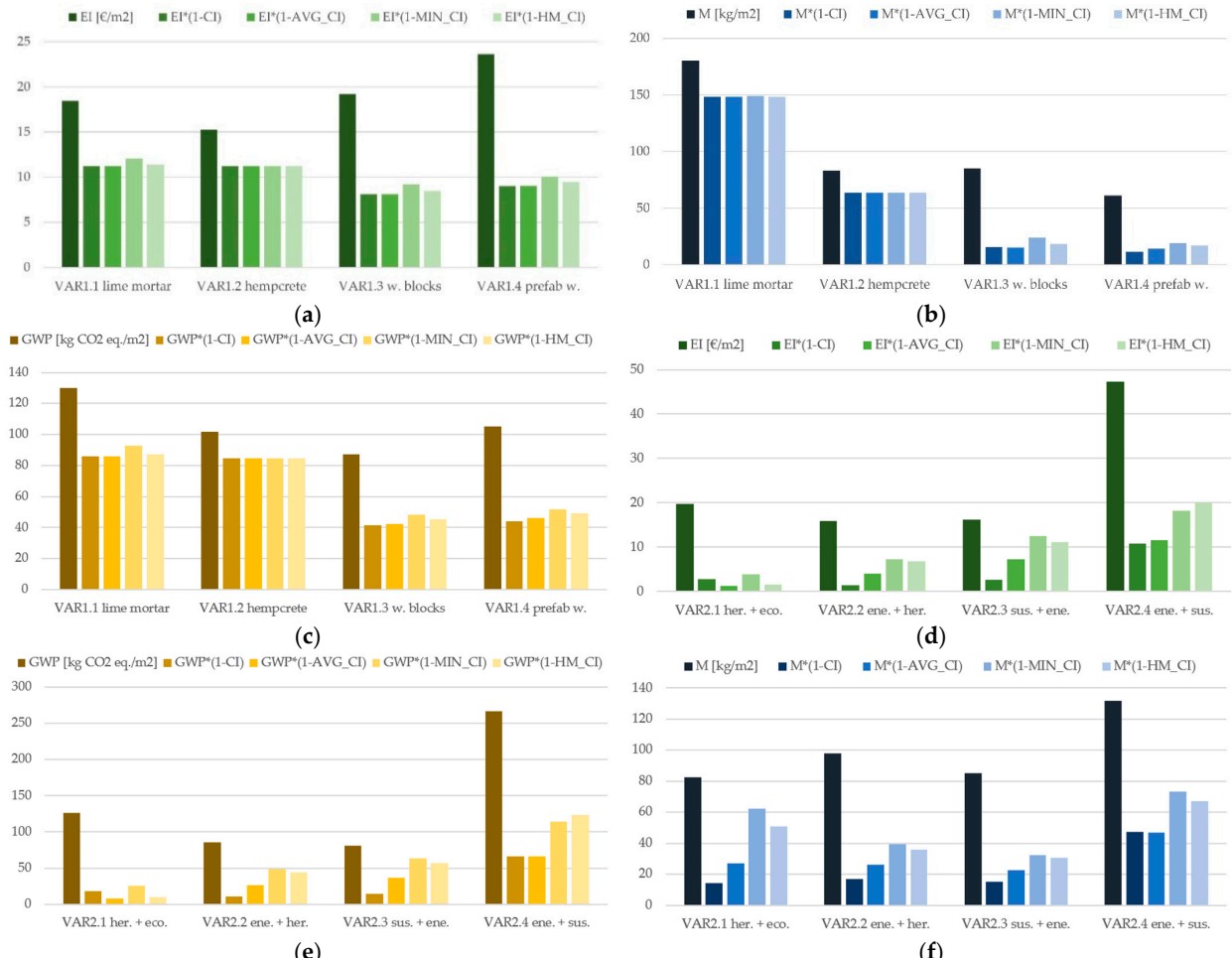

**Figure 9.** Results of the baseline EI and linked EI' (green bar charts), baseline GWP and linked GWP' (yellow bar charts), and baseline M and linked M' (blue bar charts) of all external wall variants of both case studies, grouped per external wall variant. The left bar of each external wall variant represents the baseline reference, and the remaining four bars represent the four linked results. (**a**) EI and four possible EI' of case study 1. (**b**) GWP and four possible GWP' of case study 1. (**c**) M and four possible M' of case study 1. (**d**) EI and four possible EI' of case study 2. (**e**) GWP and four possible GWP' of case study 2. (**f**) M and four possible M' of case study 2.

To see whether applying the EI, GWP or M as a variable has an effect, the difference in the EI of the variant with the highest EI and the lowest EI—and similarly for the difference in GWP and M—have been calculated per case study (see Table 8). The table shows that after linking the results, the difference decreases with the exception of the differences in GWP and M in case study 1.

**Table 8.** Difference ($\Delta$) between the variants with the highest and lowest EI, GWP or M per case study, in absolute values.

| | Case Study 1 | | | Case Study 2 | | |
| --- | --- | --- | --- | --- | --- | --- |
| | $\Delta$ EI [€/m$^2$] | $\Delta$ GWP [kg CO$_2$ eq./m$^2$] | $\Delta$ M [kg/m$^2$] | $\Delta$ EI [€/m$^2$] | $\Delta$ GWP [kg CO$_2$ eq./m$^2$] | $\Delta$ M [kg/m$^2$] |
| **Baseline unlinked results** | 8.36 | 42.69 | 118.9 | 31.36 | 185.99 | 49.4 |
| **Linked with mat_CI** | 3.11 | 44.69 | 136.7 | 9.34 | 56.22 | 33.0 |
| **Linked with AVR_CI** | 3.16 | 43.58 | 134.1 | 10.26 | 57.52 | 24.3 |
| **Linked with MIN_CI** | 2.87 | 44.63 | 130.0 | 14.19 | 88.52 | 40.9 |
| **Linked with HM_CI** | 2.88 | 42.03 | 131.5 | 18.59 | 113.76 | 36.5 |

## 4. Conclusions

The applied linking method creates the possibility to have a simple objective weighting method, in which the environmental impact and the circularity of building solutions are expressed in one single score—contributing to the solution to a significant challenge in current circular development in the construction industry. The method allows for easy comparative assessments of different design variants without the need to define several case-specific scenarios. Additionally, it combines existing assessment frameworks. We have tried to make a simple link between LCA and circularity assessments by applying only the CI. Although, in practice, often more circularity indicators are assessed, the CI is considered as the most important indicator in these cases, as it stipulates the condition as to whether a solution can be easily detached and reused.

The results show that linking LCA results or mass input flows to CI has an effect, and therefore influences the decision-making in the case of a comparative assessment. As shown in the results of case study 1, the variant with the highest EI shifts to a variant with one of the lowest EI when the EI is linked to the CI. When looking at the mass input flows, this kind of big shift does not occur with regard to case study 1. With regard to case study 2, the linking exercise gives a more varied picture, with a less straightforward possibility of taking a decision as to which variant is the most interesting one.

On the one hand, the contrast between the two case studies is caused by the difference in complexity of the component compositions. The components of the external walls in case study 1 consist of a maximum of four materials that often have a comparable CI. While in case study 2, the majority of components consists of more materials with more differences in CIs, as shown in Tables 5 and 6.

On the other hand, in the process of defining the different external wall variants, in case study 1 the focus laid on the type of connections of the load-bearing structure, while in case study 2 the design strategies determined the choice of materials.

Choosing to apply the EI, GWP or M as a variable to link to the CI does have an effect. In the case of linking the CI to the EI or GWP, the difference between the external wall variants becomes smaller for both case studies. While linking the CI to the M, the difference between the variants increases in case study 1, but decreases in case study 2. Seeing this effect, the choice between an LCA indicator or M is important in view of decision-making regarding sustainability. What causes this effect cannot be clearly deducted from the case study results, and requires more case studies. In addition, it can be questioned whether M is the most logical variable—as is done in MFA-based circularity assessments—and whether it causes more inconsistent assessment results. Also considering that, in LCA, the mass of materials is, in fact, included in the life cycle inventory assessment and impact assessment too.

This exploration also showed possibilities on how to aggregate circularity scores on material level to a score on the element level. During several circularity projects—including the ones from which the two case studies are taken—we noticed that there are no consistent guidelines yet on how to aggregate or weigh circularity scores on material/product level in order to arrive at a score on element level, let alone a score on building level. In other words, coordinated guidelines on this matter would come in useful.

## 5. Discussion

Which way of linking gives the most realistic results, cannot be concluded from the results, as case study 1 shows that the way of linking makes little difference, while case study 2 shows that the way of linking has a much bigger influence. As Van Stijn et al., mention, blending approaches could also increase complexity and cloud the (dis)advantages of each approach [29]. In future research, these theoretical calculations will have to be validated through experimenting in real life cases. Does the calculated decrease of aggregated EI or GWP indeed occur in reality? Moreover, this raises the question as to whether there would also be a significant range in the linked results between the different linking ways if the same

linking exercise would be done in another project, and whether this would indicate that there was little attention given to the type of connections while making the design variants.

What could question the objectivity of this linking method is the determination of the unitless scores of the CI that vary between 0.10 and 1.00. Further assessing the influence of these scores or increasing the considered level of detail can be part of additional research. Moreover, more research is necessary to combine the other DfD indicators with LCA and other outcomes, such as Life Cycle Costing, and each other, amongst others, based on a sensitivity analysis.

Something that was not considered in this paper is the possibility that a material could have a non-reversible connection type while on a higher level—e.g., on the element level-additional connections could exist, allowing the reuse of a complete element and not the individual materials within that element. Inconsistency between the functional unit and system boundaries leads to potential errors and increases the scope of outcomes. Considering this possibility would require additional methodological rules—which are currently under investigation—that would complicate the presented linking method by applying a certain hierarchal flow chart model and sensitivity analysis.

**Author Contributions:** Conceptualization, W.C.L., S.C. and M.R.; methodology, W.C.L., S.C. and M.R.; validation, W.C.L., S.C. and M.R.; formal analysis, W.C.L., S.C. and M.R.; writing—original draft preparation, W.C.L.; writing—review and editing, W.C.L., S.C. and M.R.; visualization, W.C.L. All authors have read and agreed to the published version of the manuscript.

**Funding:** This research has been partly conducted in the Horizon 2020 project ICEBERG, under the European Union's Horizon 2020 research and innovation program under grant agreement ID: 869336.

**Acknowledgments:** This research was made possible thanks to our partners Groep Van Roey, VUB, BBRI, and UA within the project 'Circulaire School voor de Toekomst' (funded by Vlaanderen Circulair) by contributing the information of case study 1; and thanks to our partners MAKER Architecten, Origin Architecture & Engineering, VK Architects & Engineers, JUUNOO, and Beneens within the project 'Het ontwerpen, ontwikkelen en uitvoeren van renovatiestrategieën van na-oorlogse architectuur' (funded by VUB) by contributing the information of case study 2). The authors thank Helene Claes for the proofreading.

**Conflicts of Interest:** The authors declare no conflict of interest.

## Abbreviations

| | |
|---|---|
| AC | Accessibility of the Connection |
| AVG | Average |
| BAU | Business As Usual |
| BCI | Building Circularity Index |
| CI | Connection Index |
| CPR | Construction Products Regulation |
| CTI | Circular Transition Indicators |
| DfD | Design for Disassembly |
| DI | Detachability Index |
| EC | European Commission |
| EEC | Environmental External Costs |
| EI | Environmental Impact |
| EN | European Standard |
| EOL | End-of-Life |
| eq. | equivalents |
| GE | Geometry of building solution Edges |
| GWP | Global Warming Potential |
| HM | Harmonic Mean |
| IP | Interpenetrations |
| LCA | Life Cycle Assessment |
| LCIA | Life Cycle Impact Assessment |
| M | Mass (input flows) |

| MFA | Material Flow Analysis |
| MIN | Minimum |
| PEF | Product Environmental Footprint |
| RPT | Reuse Potential Tool |
| SPI | Sustainable Products Initiative |
| TC | Type of Connection |
| TCT | Transformation Capacity Tool |
| TOTEM | Tool to Optimise the Environmental impact of Materials |

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
