# Peer review of "Exploring the Missing Link between Life Cycle Assessment and Circularity Assessment in the Built Environment"

_buildings, doi:10.3390/buildings12122152_

Round 1
Reviewer 1 Report
The paper concerns a possibility of linking LCA and a circularity indicator, in order to reduce subjectivity in circularity assessments and assumptions related to case-specific scenarios that can influence the results of comparative LCA of circular building solutions.
The study starts from a very interesting idea and above all of considerable interest and relevance, and is developed with an original and shareable methodology. The results are clear and effectively presented.
The discussion also seems to include conclusions: perhaps the future developments referred to in the last part of the article, could be more emphasized and illustrated.
Reviewer 2 Report
There are two major issues in this paper that demerit its publication.
1. Using a highly subjective method of estimation (LCA) to estimate another variable.
2. Building a link between LCA outputs on one end and circularity on other. This link is not a valid item to build the way it is being done in this paper. Slight change in inputs can result in different LCA outputs thus change in circularity. The slight change can be due to misunderstanding of assumptions, deliberate attempt to malign or upsell something.
Reviewer 3 Report
The manuscript showed an intriguing subject for researchers and practitioners who are involved in construction works. It was sufficiently well presented for this journal. Following are my comments:
Overall, writing was fine. Please, go through the proofing of the manuscript. There were expressions and syntax which can be improved.
In Section 1, a variety of governmental documents and guidelines were introduced to illustrate the treatment of environmental impacts related to construction practice in Europe. In my opinion, discussion needs to be expanded by introducing more research articles. This section needs to outline the current situations of relevant research fields such as LCA and circularity assessment. It will help readers understand the scientific significance and need of the performed research.
In this study, operational energy use (B6 in Figure 2) was excluded. It would be fine as it was justified in the manuscript. However, the case studies are located in different regions. Please make sure that they are in the same climate conditions including climate zones and solar orientations. Otherwise, operational energy use will be different if they have equal thermal performances.
Please, explain a bit more to justify the choice of buildings as case studies. In which way were they appropriate for this study? Are they representative of any sort of building typologies?
Section 4 was redundantly long. I would suggest that another section is added as conclusions. It would convey the essence of the performed research to readers better.
Round 2
Reviewer 2 Report
I understand it takes a lot of work to come up with Novel ideas, but I think this linking approach is not novel that merits a journal publication. There is a difference between journal articles and magazine articles, what is novel in this relationship? what a person can learn from it?
Author Response
We have included a new figure and the following text and additional reference in the manuscript:
'It is an exploration of integrating LCA and a circularity assessment (as visualised in Figure 1) instead of performing them in parallel like done in current practice. This study therefore attempts to bridge the gap between quantitative LCA outcomes and a dimensionless simplified circularity rating. The need to clarify the link between LCA and circularity indicators is also underlined by the set-up of a joint research group between the Society for Environmental Toxicology And Chemistry (SETAC) and the American Center for Life Cycle Assessment (ACLA) [32].
<figure 1>
Figure 1. The aim of this paper: linking LCA (left part of the figure based on EN 15804 [23]) to circularity assessment (assessing the application of circular economy strategies as visualised in the right part of the figure) in one single score without the need to define case-specific scenarios.’
[32] Saidani, M.; Kreuder, A.; Babilonia, G.; Benavides, P.T.; Blume, N.; Jackson, S.; Koffler, C.; Kumar, M.; Minke, C.; Richkus, J.; et al. Clarify the nexus between life cycle assessment and circularity indicators: a SETAC/ACLCA interest group. Int. J. Life Cycle Assess. 2022, 27, 916–925, doi:10.1007/s11367-022-02061-w.
Reviewer 3 Report
Please, highlight the modified parts so that reviewers can tell which paragraphs have been altered. If the authors mean that the entire manuscript has been rewritten, it should be regarded as a new submission.
Author Response
In the second revision of our manuscript we have tracked the changes between the second revision and the first version of 12 May, and in its reference list, the references that were added in the two revisions are marked in yellow.Round 3
Reviewer 3 Report
My comments have been addressed.
Author Response
Thanks for letting us know that we have addressed all your comments.